Monitoring the critically endangered Clanwilliam cedar with freely available Google Earth imagery

http://orcid.org/0000-0003-1246-1181 Slingsby Jasper A. 1 2 jasper@saeon.ac.za
Slingsby Peter W. O. 3
1 Fynbos Node, South African Environmental Observation Network (SAEON) , Cape Town, Western Cape , South Africa
2 Centre for Statistics in Ecology, the Environment and Conservation, Department of Biological Sciences, University of Cape Town , Cape Town, Western Cape , South Africa
3 Slingsby Maps , Cape Town, Western Cape , South Africa
Culham Alastair
Electronic publication date: 2019 Jul 3
Publication date: 2019
Volume: 7
Electronic Location ID: e7005
Received 2019 Feb 12; Accepted 2019 Apr 23
Copyright: © 2019 Slingsby and Slingsby
Copyright year: 2019
Copyright holder: Slingsby and Slingsby
License: This is an open access article distributed under the terms of the Creative Commons Attribution License, which permits unrestricted use, distribution, reproduction and adaptation in any medium and for any purpose provided that it is properly attributed. For attribution, the original author(s), title, publication source (PeerJ) and either DOI or URL of the article must be cited.
License URL: https://creativecommons.org/licenses/by/4.0/

Keywords: Widdringtonia cedarbergensis, Essential biodiversity variables (EBVs), Species and population monitoring, Machine learning, Global distribution monitoring, Widdringtonia wallichii

Funding: National Research Foundation of South Africa 118593 RReTool: Rapid and repeatable tools for monitoring and mitigating global change impacts on natural resources project This work is based on the research supported by the National Research Foundation of South Africa (Grant Number 118593) as part of the RReTool: Rapid and repeatable tools for monitoring and mitigating global change impacts on natural resources project. The funders had no role in study design, data collection and analysis, decision to publish, or preparation of the manuscript.

==============================
Monitoring of species and populations is essential for biodiversity observation and reporting at local, national and global scales, but can be an exceedingly difficult task for many, if not most, species. We tested the viability of using Google Earth™ imagery to manually map and monitor all individuals of the critically endangered Clanwilliam cedar, Widdringtonia wallichii Endl. ex Carrière, across its global native distribution; the remote and rugged Cederberg mountains. Comparison with sampling from field surveys reveals this to be a highly efficient and effective method for mapping healthy adult tree localities, but it fails to detect small or unhealthy individuals with green canopies <4 m2, or discern the number of individuals in clumps. This approach is clearly viable as a monitoring tool for this species and, with the rapid progress being made in machine learning approaches and satellite technology, will only become easier and more feasible for a greater number of species in the near future. Sadly, our field surveys revealed that the number of trees that have recently died (dead leaves still present) outnumbered live trees by a ratio of 2:1.

Introduction

The charismatic Clanwilliam cedar, Widdringtonia wallichii Endl. ex Carrière (routinely referred to by the later homotypic synonym W. cedarbergensis Marsh), a narrow endemic within the Cederberg mountains, Fynbos Biome, South Africa (32°18′S 19°06′E), has shown precipitous decline in population numbers over the past two centuries. While there are anecdotes of overexploitation in the early 1800s (Smith, 1955), recent evidence from analysis of repeat photographs suggests that mortality has been exacerbated by anthropogenic climate change, particularly over the past 30 years (White et al., 2016). This is consistent with a global analysis revealing increased climate-induced tree mortality over the past 40 years (Allen et al., 2010), with conifer species being especially vulnerable. With increasing evidence of climate change impacts on South African vegetation (Foden et al., 2007; Slingsby et al., 2017; White et al., 2016), it is key that we improve our ability to detect and track these impacts, both to raise public awareness and to improve our understanding of anticipated environmental change.

While the monitoring of species and populations is one of the six major classes of Essential Biodiversity Variables “required to study, report, and manage biodiversity change” (Pereira et al., 2013), this can be an exceedingly difficult task in rugged and remote landscapes, or where species are difficult to detect. Fortunately, freely available, high resolution satellite imagery is making this more feasible for large organisms such as trees (Visser et al., 2014; Geller et al., 2017). Here we test the viability of using Google Earth™ imagery to map and monitor all individuals of the Clanwilliam cedar across its global native distribution.

Material and Methods

Tree localities for the entire species’ distribution were manually mapped from high-resolution CNES/Airbus satellite imagery available from Google Earth™ for the year 2013. Trees were identified based on canopy color, size, shape and shadows, and, where possible, verified with ground photographs from the publicly contributed archives accessible through Google Earth™ and a personal collection of ∼19,000 georeferenced images from research for the Cederberg hiking map (Slingsby, 2015). Early tests found that we could not detect dead trees, likely because they cast very little shadow and their stems are mostly white and cannot be discerned from the high cover of white rock in the area. Trees with brown canopies were ignored as they were likely dead and/or other species. For visual identification and mapping, Google Earth™ scenes were exported to CorelDRAW® (Core, Ottawa, Canada), the color balance adjusted, and trees marked as points in a layer. The tree points layer was then exported as a vector image and georeferenced and converted to keyhole markup language (KML) in ArcGIS 10.2 (Esri, Redlands, CA, USA). Minimum horizontal mapping accuracy was established by opening the KML in Google Earth™ and measuring the distance between 200 mapped points and the trees they represent using the measuring tool. Dense areas were avoided to reduce confusion between target trees.

To validate our satellite enumeration approach on the ground, we mapped the GPS location and size class (adult = canopy >4 m2, sub-adult = canopy >1 and <4 m2, and seedlings = canopy <1 m2) of all cedar trees found within three circumscribed field sites across the species’ range. We then compared our field survey results with population estimates from our satellite image analysis, exploring the influence of size class on detection from satellite. Since our first site survey revealed that the trees can survive substantial canopy dieback, we also recorded the size of the live canopy of trees for the two subsequent field sites to explore the effect of live canopy size on detection from satellite.

Results

We mapped 13,419 cedar tree localities (Fig. 1), taking an estimated 200 working hours. None of our 200 sample trees fell more than 20 m from the mapped point, suggesting a horizontal mapping accuracy <20 m (∼ 1:24,000).

Figure 1 Clanwilliam cedar size class distribution and tree localities.

(A) Barplot of trees of different size classes (A = adult; canopy >4 m2, SA = sub-adult, canopy >1 and <4 m2, S = seedling; canopy <1 m2) within our field sites observed on the ground or using satellite imagery from Google Earth™. (B) Map of Clanwilliam cedar tree localities (black points) mapped from 2013 Google Earth™ imagery showing the Cederberg Wilderness Area boundary (dashed line), and field survey sites (white circles with black centre). Terrain image generated from the Shuttle Radar Topography Mission (SRTM) 90 m digital elevation model (Jarvis et al., 2008).

Our ground surveys took five days for a team of two to cover 1/20,000th of the Clanwilliam cedar’s range. We found 123 live trees (61 adults, 24 subadults and 38 seedlings), while our satellite approach detected only 21 healthy green canopies in the same area (Fig. 1). Our canopy health data from two of the three field sites revealed that of the 25 live adult trees only 10 had healthy green canopies >4 m2, while our satellite approach counted nine trees. Our field survey also revealed 237 dead trees (i.e., a ratio of two dead to every live tree), made up of 109 adults, 82 sub-adults and 46 seedlings, still bearing dead leaves.

Discussion

Our satellite-based approach did very well to provide a near-perfect fine-scale description of the Clanwilliam cedar’s distribution, providing a detailed baseline that allows monitoring of future change, and allowing inference of fine-scale habitat preferences that could lead to a better understanding of the species’ ecology and causes of its decline. While the satellite image analysis clearly missed smaller individuals and those with unhealthy canopies, and cannot discern between clumps of trees and single individuals, it provides a very good indication of the locations of adult trees with live canopies. We achieved a horizontal mapping accuracy suitable for most applications, but it could likely be improved if all analyses were performed directly in Google Earth™ or geographic information system. This would likely require the ability to modify the color balance of images directly in the software to aid visual detection. With good field estimates of the species’ size class distribution and canopy health it would be feasible to provide a relatively accurate estimate of population numbers based on the locality data. Our small field survey and work by White et al. (2016) both suggest that population structure, canopy health, recruitment (presence of seedlings) and mortality (presence of dead stems) are highly varied across the Cederberg, cautioning against extrapolation without sufficient sampling, stratified across environmental gradients and spanning the species’ full range.

There was no evidence to suggest there were any errors of commission, whereby individuals of other species were mistaken for the Clanwilliam cedar. The most likely species would have been Heeria argentea (Thunb.) Meisn. or Podocarpus elongatus (Aiton) L’Hér. ex Pers., but these were readily distinguishable by differences in canopy color, shape, shadow and habitat. Omission rates may vary depending on topography and the recent occurrence of fire; but error rates for localities with adult trees or clumps >4 m2 are likely to be low.

This observation method is clearly highly efficient and effective, and has great potential for application to other important plant species worldwide, especially large trees or shrubs that occur in sparse vegetation. Key species in South Africa include the declining Aloidendron dichotomum (Masson) Klopper & Gideon (Foden et al., 2007), large species in the Proteaceae Juss. (Schurr et al., 2012), or savanna trees.

While our approach is far cheaper and more time-efficient than an exhaustive (and exhausting!) field survey, the field of image analysis with machine learning approaches is moving incredibly rapidly (Demir et al., 2018) and will greatly reduce the need for and time spent doing manual digitization of individual localities. This, combined with the growing record of satellite and aerial imagery with continually improving spatial and spectral resolution will soon allow for rapid and cost effective monitoring of many species across their global distribution ranges (Geller et al., 2017).

The authors would like to thank Google for making their imagery freely available. We also thank Amy Slingsby, Glenn Moncrieff, Nicky Allsopp and Abri de Buys for assistance with field work, and Thomas Slingsby for converting the tree layer into Keyhole Markup Language (KML) format.

Additional Information and Declarations

Competing Interests

Author Contributions

Data Availability

Jasper A. Slingsby declares no competing interests. Peter W. O. Slingsby is employed by Slingsby Maps.

Jasper A. Slingsby conceived and designed the experiments, performed the experiments, analyzed the data, contributed reagents/materials/analysis tools, prepared figures and/or tables, authored or reviewed drafts of the paper, approved the final draft.

Peter W. O. Slingsby conceived and designed the experiments, performed the experiments, contributed reagents/materials/analysis tools, authored or reviewed drafts of the paper, approved the final draft.

The following information was supplied regarding data availability:

The field survey are available as a GeoPackage and the 13,419 cedar tree localities are available in Keyhole Markup Language (KML) format via Figshare: Slingsby, Jasper; Slingsby, Peter (2019): Global distribution of Clanwilliam cedar tree localities in 2013. figshare. Dataset. DOI 10.6084/m9.figshare.7670435.v1.

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
