# Peer review of "Monitoring the critically endangered Clanwilliam cedar with freely available Google Earth imagery"

_PeerJ, doi:10.7717/peerj.7005_

## Round 0.1 · original submission · Minor Revisions

Your paper was considered of interest and relevance by both reviewers however it is clear that you need to improve in the reporting of your methods and results. While there are many edits to make, none of these should require substantial work so I have treated these as minor revsions.

·

Basic reporting

Clear and unambiguous, professional English used throughout:
This is a very well written and coherent paper with almost no English language errors. The few I found are listed below:
“Fynbos Biome” on line 37 should be lower case.
“6” on line 47 should be written as “six”.
Line 88: “satellite based” should be “satellite-based”
Line 89: "Clanwilliam Cedar’s” should be “Clanwilliam cedar’s”
Line 105: “from other species” should be “of other species”.


Literature references, sufficient field background/context provided:
I thought the introduction was succinct, but well contextualised.


Professional article structure, figures, tables. Raw data shared:
The paper conforms to the journal’s requirements with respect to structure.

The one figure also largely conforms to the journal requirements, except needs uppercase labels for the subfigures.

The raw data are made available via Figshare.


Self-contained with relevant results to hypotheses:
The paper represents a coherent, self-contained body of work.

Experimental design

Original primary research within Aims and Scope of the journal.
The research question of this paper, testing the viability of using Google Earth imagery to map and monitor individuals of the Clanwilliam cedar, is clearly defined and very relevant to, among others, conservationists, ecologists and remote sensing scientists.


Rigorous investigation performed to a high technical & ethical standard.
The investigation is highly rigorous.


Methods described with sufficient detail & information to replicate.
In lines 59-60 you state that tree identifications were “verified with ground photographs from a publicly contributed archive accessible through Google EarthTM.” Would it be possible to provide more details on this archive? For example, are you referring to the “Google Maps Photos” and/or “360 Cities” layers in Google Earth?

Could you provide more details of the “three circumscribed field sites” (line 28) used to validate your “satellite enumeration approach”? Perhaps a table with coordinates, or even better a KML file with the boundaries of these sites.

Validity of the findings

Impact and novelty not assessed. Negative/inconclusive results accepted. Meaningful replication encouraged where rationale & benefit to literature is clearly stated.
No issues in this regard.


Data is robust, statistically sound, & controlled.
The data seem to be robust. However, I would like to see a few more details reported to support some of the conclusions:

Line 77: Did you get horizontal mapping accuracy for each individual/clump identified in the satellite imagery? If so, the mean and variability in horizontal mapping accuracy should be reported.

Line 79-80 (and actually 82-84): Can you provide some statistics together with these numbers? For example, what was the total accuracy? Ommission error? Can you make it clear that none of the dead trees were detected in the Google Earth imagery (lines 82-84).

Linked to the above, can you provide an indication of the number of clumps of individuals that were identified as individual trees using the satellite imagery (otherwise your results do not support your conclusion in lines 92-93 (“and cannot discern between clumps of trees and single individuals”).

Although the sample sizes are very small, it would be helpful to report the results per field site (perhaps as a supplementary table, or by subdiving the bars in Fig. 1A to represent the three field sites). This is especially important to be able to support your statement in lines 99-103: “Our small field survey and work by White et al. (2016) both suggest that population structure, canopy health, recruitment (presence of seedlings) and mortality (presence of dead stems) are highly varied across the Cederberg, cautioning against extrapolation without sufficient sampling, stratified across environmental gradients and spanning the species’ full range”.


Conclusions are well stated, linked to original research question & limited to supporting results.
In general the conclusions were very clearly stated and linked to the original research question. I think the opening sentence of the discussion, “Our satellite based approach did very well to provide a near-perfect fine-scale description of the Clanwilliam Cedar’s distribution...” will be better supported once the above changes are made to the results.
I felt there were only two other statements that need further clarification of the results in order for these to be robustly supported, as described in the previous section:
Lines 92-93
Lines 99-103
I also think that lines 199-122, although undoubtedly true, could use a reference to support this statement.


Speculation is welcome, but should be identified as such.
I felt the speculation in the discussion was reasonable and well supported. For example, in lines 97-99, the authors state “With good field estimates of the species’ size class distribution and canopy health it would be feasible to provide a relatively accurate estimate of population numbers based on the locality data”.

In the lines following the above statement, lines 99-103, I have already highlighted that I feel the authors need to provide a breakdown of the results by each of the three field sites in order to support their suggestion that “population structure, canopy health, recruitment (presence of seedlings) and mortality (presence of dead stems) are highly varied across the Cederberg...”

Additional comments

Widdringtonia cedarbergensis Marsh is now recognised as Widdringtonia wallichii Endl. ex Carrière (e.g. http://www.theplantlist.org/tpl1.1/record/kew-2466623)

I wonder if it would be helpful to have an additional few subfigures in Figure 1 where each of the field sites are shown together with the trees identified in the field, and those identified using Google Earth highlighted in another colour / point shape.

·

Basic reporting

Slingsby & Slingsby present an original approach to mapping critically endangered Widdringtonia cedarbergensis trees. The paper provides an interesting case study for the manual monitoring of large trees using open access satellite imagery and the fine-scale location data provided is of great value for conservation, research and teaching purposes.

The paper is generally well written and structured in professional English and reads well throughout. Figure 1a is relevant and well described, though Figure 1b needs both axes labels and a scale bar and could also possibly benefit from an inset to show regional context and if enlarged as a separate figure, the 3 field sites used could be added on.

The introduction could benefit from additional context on whether similar methods have been used before or the novelty of this approach to monitoring large organisms, such as trees.

The raw kml file was successfully downloaded from figshare.

Experimental design

The methods need further detail and information. Specifically, the method validation and ground surveys need clarification. I feel a reference or explanation of horizontal mapping accuracy as a validation method is necessary. It is unclear what constitutes a healthy or unhealthy tree relative to percentage canopy cover.

Validity of the findings

The results needs more detail on the method validation, specifically presenting data on the horizontal mapping accuracy comparisons. I am not experienced in mapping applications, but I assume each sampled tree has an accuracy value in m, meaning a mean ± sd could be reported.

Additional comments

The strength of this paper is the excellent fine-scale, healthy adult, location dataset produced from a relatively efficient and low-cost method. As it currently stands, the method validation has not been clearly demonstrated and requires clarification from the authors.

Specific comments below:

L2. Consider changing ‘from space’ to ‘using high resolution satellite imagery’

L57. Resolution in m/arcsec?

L57. Add a note on the spatial extent of the cedar ‘search’. Only within the Cederberg WA or within suspected historical extent?

L71. “Explore the effect of canopy health.” On what? Satellite detection?

L77. This seems like one of the most important metrics in the paper, yet was not explained in the methods. See previous comments.

L78. This is an arbitrary scale – change to km/km2 covered.

L80. The method identified 21/61 adults when considering all ground surveyed sites, but 9/10 from two of the three sites. Could you explain the discrepency here? Were the trees at the site not included in the canopy health survey particularly unhealthy, with lots of adults with sparse canopies?

L81. I’m unsure why canopy health was included in the analysis, as it is only measured at 2/3 sites and it is not clear how it relates to the original objectives of the paper.

---

## Round 0.2 · Minor Revisions

Both reviewers are agreed that you have made the changes they requested. In response to the query over the correct name for this species I must tell you that the original review was correct, the legitimate name for the species is Widdringtonia wallichii. Until a case is proposed and accepted for conservation of W. cedarbergensis this remains a later homotypic synonym and therefore a superfluous name. I've discussed this with the staff at WCSP. I would suggest that an acceptable workaround would be to use clanwilliam cedar consistently throught the paper and to say this is currently correctly called Widdringtonia wallichii but routinely referred to as W. cedarbergensis, a later name. That way you can be nomenclaturally correct but also engage with users of the superfluous name. If you are able to engage with constructively this I see no need for further review.

·

Basic reporting

I am happy with the changes made by the authors

Experimental design

I am happy with the changes made by the authors

Validity of the findings

I am happy with the changes made by the authors

Additional comments

I am happy with the changes made by the authors.

On the issue of the scientific name of the Clanwilliam cedar, I am not a taxonomic expert, so would be loathe to make a recommendation in this regard. I think it would be helpful that this issue is clarified with IPNI so that further taxonomic confusion does not ensue. I see that Marsh (1966; Bothalia) does not recognise Widdringtonia wallichii as the accepted name, giving it the assignation of "nom. nud."

Nomen nudum refers to a species "without a description or diagnosis or reference to a description or diagnosis" (https://www.iapt-taxon.org/icbn/frameset/0120AppendixVII.htm). However, the reference for W. wallichii does seem to include a description: https://www.biodiversitylibrary.org/item/106503#page/78/mode/1up (page 62)
This would suggest to me that perhaps Marsh (1966) was incorrect in assigning W. wallichii as "nom. nud.", and because this name is the older of the two potential names, it should be given preference.

However, as I said, I would suggest that this be taken up with a taxonomic expert and, if IPNI and The Plant List need to be modified, these changes be recommended to them.

·

Basic reporting

No comment.

Experimental design

No comment.

Validity of the findings

No comment.

Additional comments

I feel the authors have adequately addressed the issues raised by both reviewers. I can now recommend this manuscript for publication and I look forward to seeing its impact in the field.

---

## Round 0.3 · accepted · Accept

Many thanks for making those changes. Apologies for the slight delay in handling this over Easter.

#